# Ultrafast-nonlinear ultraviolet pulse modulation in an AlInGaN polariton waveguide operating up to room temperature

D. M. Di Paola[1], P. M. Walker [1✉], R. P. A. Emmanuele[1], A. V. Yulin[2], J. Ciers [3,4], Z. Zaidi[1], J.-F. Carlin[3], N. Grandjean[3], I. Shelykh[2,5], M. S. Skolnick[1,2], R. Butté [3✉] & D. N. Krizhanovskii [1,2]

Ultrafast nonlinear photonics enables a host of applications in advanced on-chip spectroscopy and information processing. These rely on a strong intensity dependent (nonlinear) refractive index capable of modulating optical pulses on sub-picosecond timescales and on length scales suitable for integrated photonics. Currently there is no platform that can provide this for the UV spectral range where broadband spectra generated by nonlinear modulation can pave the way to new on-chip ultrafast (bio-) chemical spectroscopy devices. We demonstrate the giant nonlinearity of UV hybrid light-matter states (exciton-polaritons) up to room temperature in an AlInGaN waveguide. We experimentally measure ultrafast nonlinear spectral broadening of UV pulses in a compact 100 $\mu$m long device and deduce a nonlinearity 1000 times that in common UV nonlinear materials and comparable to non-UV polariton devices. Our demonstration promises to underpin a new generation of integrated UV nonlinear light sources for advanced spectroscopy and measurement.

[1] Department of Physics and Astronomy, University of Sheffield, Sheffield, UK. [2] Department of Physics, ITMO University, St Petersburg, Russia. [3] Institute of Physics, École Polytechnique Fédérale de Lausanne (EPFL), Lausanne, Switzerland. [4] Department of Microtechnology and Nanoscience, Chalmers University of Technology, Gothenburg, Sweden. [5] Science Institute, University of Iceland, Reykjavik, Iceland. ✉email: p.m.walker@sheffield.ac.uk; raphael.butte@epfl.ch

Third order optical nonlinearities can be used to modulate the temporal envelope of optical pulses through such fundamental effects as self- and cross- phase-modulation (SPM and XPM), four wave mixing, modulational instability, soliton propagation[1], and dispersive wave emission[2]. These nonlinearities underpin many important functionalities in photonics. Nonlinear pulse compression[3,4] allows generation of ultra-short pulses. Microresonator Kerr frequency combs[5] can provide high repetition rate pulse trains starting from continuous wave (CW) light, and supercontinuum generation provides light sources with octave spanning spectra[4,6]. Further applications include amplification through four wave mixing, quantum noise squeezing[7] and all-optical logic[8]. Such nonlinearities are important in the UV spectral range because many atomic and molecular optical transitions occur there. For example, ultra-short UV pulses have been used in femtosecond studies of chemical reactions[9] or direct observation of molecular excited states in the time domain[10,11]. Broadband UV pulses can be used in absorption[12–14] or coherent Raman[15] spectroscopies. UV wavelengths are also important for Doppler cooling[16] or addressing trapped ions[17–19], or as pumps for quantum light sources[20].

While nonlinear effects are well developed in optical fibres[1,2] and bulk materials[3,21], recent advances raise the prospect of on-chip integrated nonlinear devices[5,22,23]. This can allow the miniaturisation of complicated optical systems. Large magnitude nonlinearities are required to minimise power requirements and device sizes. Unfortunately, materials commonly used for UV nonlinear optics, such as fused silica, calcium fluoride and others (see Table 1) have very small nonlinearity. One promising method to achieve large nonlinearities is to strongly couple optical modes to the excitonic resonances of quantum wells (QWs) embedded in the photonic structure. This strong coupling results in the formation of exciton-polaritons (polaritons), which are part-photon-part-exciton quasi-particles[24–27]. Their exciton-like interactions lead to nonlinearity at least 1000 times larger than in weakly coupled semiconductors[28,29]. This has allowed observation of nonlinear phenomena such as parametric scattering[30], superfluidity[31], solitons[32] and optical continuum generation[33] as well as quantum light sources[7,34]. Achieving strong coupling nonlinearity requires exciton binding energy greater than the thermal energy $k_BT$. Historically this restricted polariton devices in GaAs-based semiconductors to liquid helium temperature operation. More recently, wide bandgap semiconductors such as GaN have allowed observation of polaritons at room temperature[24,35]. Nevertheless, polariton nonlinear pulse modulation effects have not been observed at room temperature or in the UV. UV pulse nonlinear modulation has also not been

observed in AlInGaN-based devices, despite the importance of AlInGaN for this spectral range.

In this paper, we demonstrate the temporal modulation of UV optical pulses using polariton nonlinearity. The linear properties of our AlInGaN QW polariton waveguide structure were previously discussed in detail in ref. [35]. Here we show that the strong light-matter coupling provides high UV nonlinearity up to room temperature. Strong modulation of picosecond optical pulses is achieved over a short 100 μm distance leading to spectral broadening up to 80 meV width. This broadening, which cannot occur in the linear regime, is the essential signature of pulse temporal envelope modulation[33]. We deduce an effective nonlinear refractive index $> 10^{-13}$ cm$^2$ W$^{-1}$, three orders of magnitude larger than those measured in other materials commonly used for ultrafast nonlinear optics in the UV (see Table 1). The AlInGaN material system we use is highly robust and a mature and leading semiconductor technology for opto-electronics, with epitaxially grown wafers available commercially and strong excitonic optical transitions up to room temperature in the UV spectral range[36,37]. The nonlinear exciton interactions in our system are comparable to those in other polariton material systems, such as GaAs and perovskites[27] which, however, do not operate in the UV up to room temperature. Our combination of high nonlinearity and room temperature operation in an on-chip device in a mature material system shows the great potential of AlInGaN-based polaritons for ultrafast nonlinear optics in the technologically important UV spectral range.

## Results

**Device properties.** A schematic of our structure is shown in Fig. 1a. It is a planar waveguide with a core containing 22 GaN/ Al$_{0.1}$Ga$_{0.9}$N QWs grown on a lattice matched AlInN cladding layer on a low dislocation density (~$10^6$ cm$^{-2}$) freestanding GaN

| Table 1 Nonlinear refractive indexes of materials commonly used for nonlinear optics with UV pulses. | | |
|---|---|---|
| **Material** | **$n_2$ ($10^{-16}$ cm$^2$ W$^{-1}$)** | **Wavelength (nm)** |
| This work | ~1900–4000 | 353 |
| Bulk GaN[44,45] | ~100 | 527 |
| AlN[53] | 35 | 1558 |
| Silicon nitride[54] | 26 | 1550 |
| Diamond[55] | 14 | 354 |
| Yttrium orthosilicate[23] | 6 | 800 |
| Al$_2$O$_3$[56] | 3.7 | 355 |
| BBO[56] | 3.6 | 355 |
| BaF$_2$[21] | 2.7 | 355 |
| Fused silica[56] | 2.4 | 355 |
| CaF$_2$[21] | 1.92 | 308 |
| LiF[56] | 0.6 | 355 |
| MgF$_2$[56] | 0.6 | 355 |

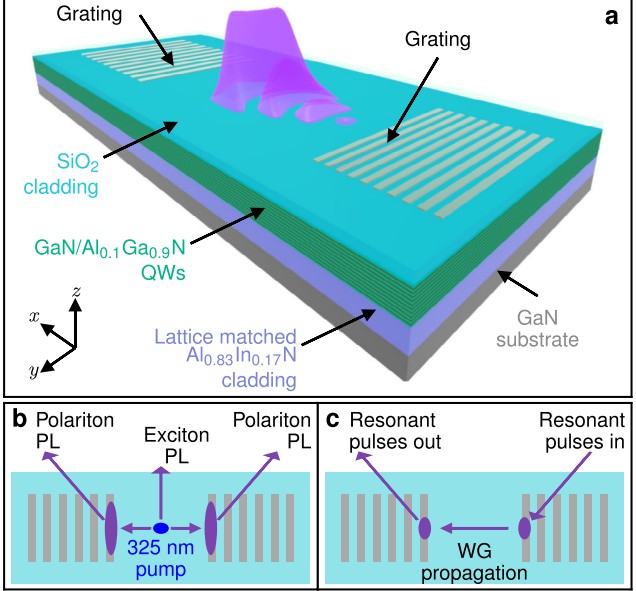

**Fig. 1 Sample and experimental schematics. a** Schematic of AlInGaN polariton waveguide (WG) structure. **b** Excitation and detection schematic for photoluminescence (PL) measurements. Hot carriers are excited by 325 nm laser light. Polariton states are populated by luminescence, propagate to the grating couplers, and are diffracted out. **c** Excitation and detection schematic for nonlinear measurements. Pulses are resonantly coupled to the waveguide mode at one grating coupler, propagate in an unpatterned region between the couplers, and are diffracted out at a second coupler.

substrate[35]. Nickel grating couplers were fabricated directly on top of the core and the whole structure was then capped with a silicon dioxide cladding. Details of the layer refractive indexes and optical guided modes are given in Supplementary Fig. 1 and Supplementary Discussion 1. Apart from the grating couplers the sample structure and linear properties are nominally identical to those previously reported in ref. [35]. The gratings function to diffract light from the waveguide guided modes, which propagate close to the x-axis in the x–y plane, to free space modes at angles close to the z-axis. In Fig. 1b we depict incoherent waveguide polaritons generated through photoluminescence (PL) propagating to the gratings and being diffracted out to be detected. Alternatively, the gratings can be used to directly couple incident laser beams into the guided modes. Thus we can inject optical pulses into the waveguide at one grating, allow them to propagate in a region between two gratings, and then couple them back out to free space at a second grating (see Fig. 1c). Nonlinear processes may occur during the propagation of these coherent laser pulses inside the waveguide.

**Photoluminescence measurements**. We first characterise the basic properties of our sample using photoluminescence (PL) measurements. For these measurements, the excitation was at a spatial position between two gratings and at a wavelength of 325 nm, much higher in energy than the QW band edge (see Fig. 1b). Hot carrier relaxation populates all exciton and polariton states. We detect exciton PL from the excitation spot and polariton PL which has propagated to the gratings. See methods for more details. Figure 2a shows the exciton PL spectra for several sample temperatures. The spectral features agree well with those already reported for the nominally identical sample in ref. [35], which confirms that our sample has the same electronic properties. Figure 2b shows the polariton PL after propagation to the gratings. The emission peaks ~35 meV and 92 meV below the A exciton again agree well with ref. [35]. The total spectrum is ~75 meV wide, which is expected since hot carrier relaxation incoherently populates all available lower energy states. We note that there is no significant difference between the two curves at 4 K which were excited with an order of magnitude difference in power. This is expected since the incoherent nature of PL precludes nonlinear-optical spectral broadening mechanisms. More detailed discussion of the PL data is given in Supplementary Discussion 2.

To confirm that the waveguide optical modes couple strongly to the QW excitons, as in ref. [35], we also performed angle-resolved PL measurements. Details of this experimental technique are given in Supplementary Discussion 3. The angle-resolved spectra are shown for temperatures T = 4 K and 300 K in Fig. 2c, d. Polaritons with different waveguide wavenumbers are diffracted out to different angles by the grating couplers. The intensity peaks (seen in the colour scale) reveal the polariton dispersion relation. The spectra are symmetric around zero angle since we detect light which has travelled in both forward and backward direction from the excitation spot (see Fig. 1b). The curves LPB₁ and LPB₂ correspond to the forward and backward propagating lower polariton branch modes respectively. The dashed vertical line X is at the A exciton frequency. The dashed curve Γ shows the dispersion of the waveguide photon modes in the absence of strong coupling. These latter were calculated using the freely available CAMFR eigenmode solver (see Supplementary Discussion 1 for details). LPB₁ and LPB₂ show a clear anti-crossing between the exciton and photon and are well fit by a coupled oscillator model (solid white lines). In the fitting we account for the photonic mode dispersion and find Rabi splitting $91 \pm 4$ and $70 \pm 20$ meV at 4 K and 300 K, respectively,

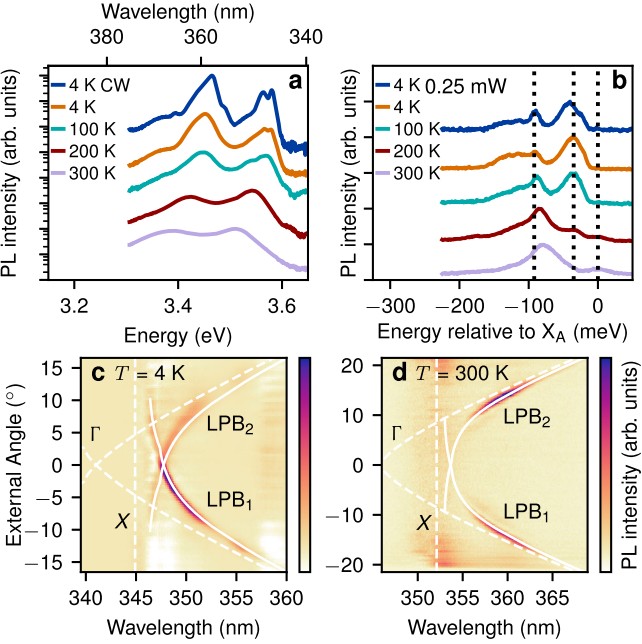

**Fig. 2 Photoluminescence (PL) characterisation. a** PL spectra emitted inside the light cone (near zero in-plane momentum) showing exciton and buffer layer emission for various temperatures. Curves are vertically offset for visibility and plotted on a logarithmic scale. CW denotes continuous-wave excitation. **b** Polariton PL spectra after propagating 50 μm from the excitation spot to the gratings (see Fig. 1b). Vertical black dashed lines are at the QW A exciton frequency and 35 meV and 92 meV below the A exciton frequency. **c, d** Angle-resolved polariton PL spectra at temperatures T = 4 K (**b**) and T = 300 K (**c**). The solid white curves (LPB₁ and LPB₂) denote the best fit forward and backward propagating lower polariton branches. The dashed white curves (Γ) give the calculated uncoupled photon dispersion. The dashed vertical line (X) denotes the exciton frequency. The exciton emission has been subtracted to highlight the polariton modes (see 'Methods').

demonstrating that the system is strongly coupled up to room temperature. Further details of the fitting are given in Supplementary Discussion 3. In summary, the linear properties of the sample agree closely with those already presented in ref. [35]. The waveguide photons and QW excitons are strongly coupled, and the polariton PL spectra are broad and show no strong power dependence.

**Pulse nonlinear self-modulation**. In order to study the nonlinear properties of the waveguide, we injected laser pulses at frequencies corresponding to the LPB directly into the guided mode through a grating coupler and detected them after 100 μm propagation (see Fig. 1c, 'Methods', and Supplementary Discussion 4). In contrast to the PL presented earlier, the light injected in this resonant way has the coherence of the laser pulses and propagates as polariton pulses inside the waveguide. This coherence is what allows the pulses to undergo nonlinear modulation. The colour maps in Fig. 3a, b show the spectrum of the pulses at the output for T = 10 K and 200 K, respectively, for increasing pulse energy coupled into the waveguide. The intensity is plotted vs. wavelength λ and the spatial position y transverse to the propagation direction. At the lowest pulse energy the unmodulated transmitted spectrum can be seen as a red peak in the region |y| < 3 μm, surrounded by a green background corresponding to a few percent of the peak and extending out to large y. This background comes from scatter of the incoming UV laser beam from the optics and is not related to the light transmitted

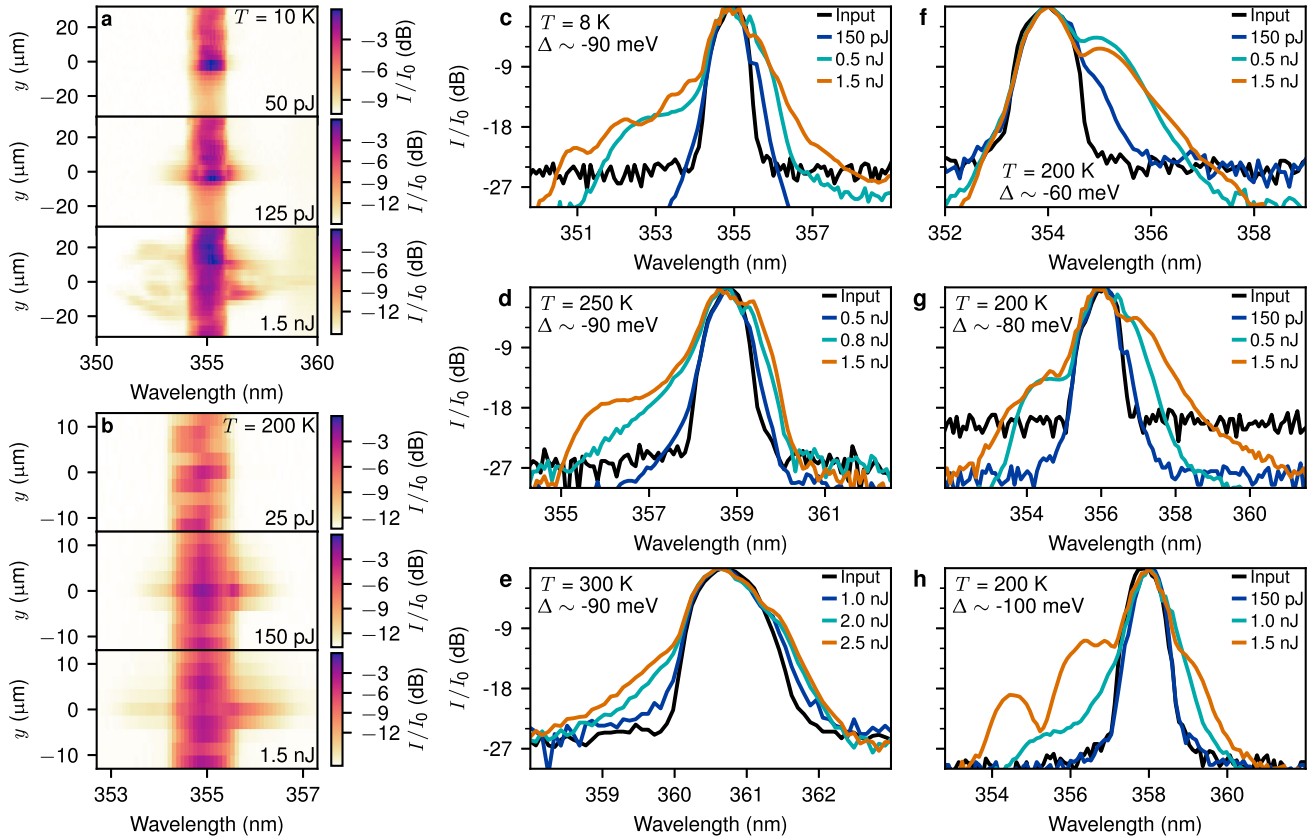

**Fig. 3 Spectra after nonlinear pulse propagation. a, b** Colour maps of the output intensity $I$ relative to the peak $I_0$ vs. wavelength, $\lambda$, and position $y$ transverse to the propagation direction for increasing pulse energy coupled into the waveguide and at temperatures $T = 10$ K (**a**) and at $T = 200$ K (**b**). The injected pulses had central wavelength $\lambda = 355$ nm and propagation distance $L = 100$ μm. **c–e** Spectra integrated along $y$ for a range of temperatures, $T = 8$ K (**c**), 250 K (**d**), and 300 K (**e**) all at pulse-exciton detuning $\Delta \sim -90$ meV. **f–h** Spectra integrated along the $y$-direction for a range of detunings $\Delta \sim -60$ meV (**f**), $-80$ meV (**g**), and $-100$ meV (**h**) all at $T = 200$ K.

through the waveguide. As the energy of the injected laser pulses increases, the spectra broaden both in wavelength $\lambda$ and along $y$, resulting in spectra with a complex inter-dependence of $y$ vs. $\lambda$ at the highest powers. We note that the spectral shape of the background scatter remains constant. As we will later confirm by comparison with simulations, the broadening of the waveguided light arises due to simultaneous nonlinear modulation of the pulse temporal and spatial ($y$) envelope[33]. The large spectral width and non-trivial $y(\lambda)$ dependence imply an optical field with features that vary rapidly, on a timescale equal to the inverse of the spectral width, which can only be produced by sub-picosecond nonlinear dynamics. The spectra we observe here are qualitatively very different to the PL spectra in Fig. 2. At low power the spectrum is narrow, corresponding to that of the incident laser, and it then broadens strongly with increasing power. The broadening always begins symmetrically about the incident laser spectrum for a wide range of laser detunings from the exciton.

In Fig. 3c–h we explore the overall broadening in $\lambda$ for a range of parameters. Figure 3c–e show the $y$-integrated spectra for a wide range of temperatures $T = 8$–300 K and for a constant detuning $\Delta \sim -90$ meV of the laser pulses from exciton frequency. With increasing pulse energy we observe spectral broadening over the whole temperature range with spectral widths at the highest measured powers of 58 meV for $T = 8$ K, 45 meV for 250 K, and 29 meV for 300 K (Fig. 3c–e), all measured at $-20$ dB from the peak. These compare to initial pulse widths of less than 16 meV. At other detunings we also achieved spectral

broadening up to 80 meV at $T = 8$ K and 66 meV at $T = 100$ K (see Supplementary Fig. 2).

In Fig. 3f–h we show the integrated spectra at $T = 200$ K for three different detunings $\Delta$. For the smallest detuning of $-60$ meV (Fig. 3f), the spectra are broadened asymmetrically with stronger broadening on the long $\lambda$ side of the pulse peak. We attribute this asymmetry to the strong absorption of wavelengths on the short $\lambda$ side which are very close to the exciton. When the detuning is increased to $\Delta \sim -80$ meV (Fig. 3g) the spectra broaden on both sides of the peak with a slight asymmetry between the short and long wavelength sidebands. Finally, at $\Delta \sim -100$ meV (Fig. 3h), the broadening is strong on the short wavelength side but weak for the long wavelengths which are further from the exciton resonance. This kind of asymmetric broadening is known to arise from a frequency-dependent nonlinearity[38]. These observations show that the nonlinearity decreases strongly as the frequency becomes further detuned from the exciton resonance at longer wavelengths. This occurs on the scale of a few tens of meV, comparable to the Rabi splitting, which is expected for a nonlinearity arising from the strong photon-exciton coupling[33].

**Comparison with numerical simulations.** The experimental spectra are in agreement with numerical simulations of propagating polaritons, shown in Fig. 4a for parameters corresponding to $T = 100$ K, $\Delta = -92$ meV and incident pulse energies 200 pJ (black lines) and 750 pJ (blue lines). We found good agreement between the widths of the numerical and experimental spectra for

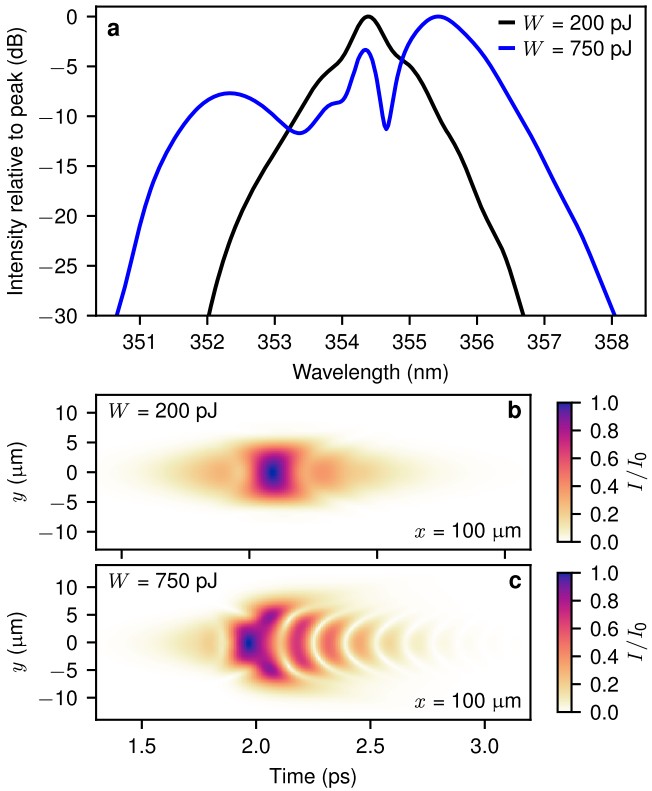

**Fig. 4 Simulated nonlinear pulse propagation. a** Numerically calculated normalized spectra of the output field corresponding to temperature $T = 100$ K, detuning $\Delta = -92$ meV and pulse energies $W$ of 200 pJ (black lines) and 750 pJ (blue). **b, c** The numerically calculated spatio-temporal distributions of the intensity $I$ of the field relative to the peak $I_0$ after 100 μm propagation in the planar waveguide for incident pulse energies 200 pJ (**b**) and 750 pJ (**c**).

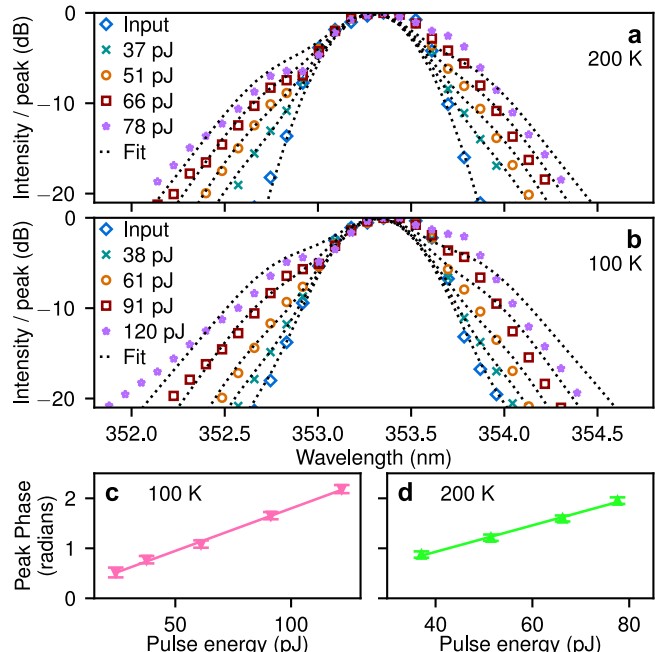

**Fig. 5 Spectral broadening in the low power limit. a, b** Comparison of experimental spectra (points) and best fit of SPM model spectra (dotted lines) for two temperatures and several pulse energies. **c, d** Peak phase from the SPM fits in (**b**) and (**a**). Error bars give the uncertainty in peak phase obtained from the least-squares fitting procedure (±2 standard deviations). Solid lines show the linear best fit.

pulse energies up to 225 pJ (see Supplementary Discussion 5 and Supplementary Figs. 3–5 for further details). The spatio-temporal distributions of the field intensity corresponding to the spectra in Fig. 4a are shown in Fig. 4b, c. As the pulse energy is increased from Fig. 4b, c we observe an increasing modulation of the field intensity coinciding with increasing spectral broadening, confirming that the spectral broadening arises from the sub-picosecond nonlinear modulation. At pulse energies higher than 225 pJ the simulations continue to show this qualitative trend although the spectral broadening no longer agrees quantitatively with the experiment. There are a number of possible reasons for this disagreement. We use a simple model of a photonic mode coupled to a single excitonic oscillator, which is the dominant electronic excitation in the system for the relevant frequencies and polarisation. However, other aspects of semiconductor physics may start to play a role at the highest intensities. These include coupling to optical and acoustic phonon modes, strong frequency-dependent absorption due to higher exciton states and the continuum above the QW band edge, and possible interactions with dark excitons and excitons at momenta far from the guided mode. We also note that at the highest powers the spectral width ~50 meV is about 1.4% of the carrier frequency so that there might be an accumulation of numerical errors due to slight departure from the slowly varying amplitude approximation.

**Self-phase-modulation**. We deduce the strength of the nonlinearity by comparing the lowest pulse energy experimental spectra to a self-phase-modulation (SPM) model[1]. A light pulse travelling through a nonlinear medium accumulates a nonlinear phase which

leads to a characteristically broadened spectrum (see Supplementary Discussion 6). To extract the nonlinear phase $\delta\phi$ at the peak of the pulse as a function of pulse energy we performed a best fit of modelled SPM-broadened spectra to the measured spectra, shown in Fig. 5a, b as dotted lines and points respectively for two temperatures. Good agreement is achieved over a range of pulse energies. We plot the obtained $\delta\phi$ vs. pulse energy in Fig. 5c, d. The points lie on a straight line within the uncertainty, as expected for an SPM mechanism, which confirms the validity of our approach. From the slope we deduce an effective nonlinear refractive index $n_2 = 1.9 \pm 0.3 \times 10^{-13}$ cm$^2$ W$^{-1}$ for 100 K and $n_2 = 3.7 \pm 1.0 \times 10^{-13}$ cm$^2$ W$^{-1}$ for 200 K (see 'Methods'). The pulse energies inside the waveguide were carefully calibrated using a combination of measured output vs. incident power and FDTD simulation of grating coupler efficiency. See 'Methods' and Supplementary Discussion 7 for more details. Our quoted uncertainty accounts for the random errors in the fitting, coupling efficiency, waveguide losses and all other parameters entering into the model. Our measured values are three orders of magnitude larger than those in other materials commonly used for UV nonlinear optics (see Table 1).

**Comparison with nonlinearity from microscopic exciton properties**. It is also possible to estimate $n_2$ in our polariton waveguide from first principles calculations based on the microscopic properties of QW excitons. Following the discussion in ref. [39] there are two microscopic mechanisms which can contribute to the polariton nonlinearity. Higher pulse powers correspond to higher numbers of excitons per unit area (exciton density) in the QWs. Repulsive interactions cause the exciton resonance frequency to increase (blueshift) which then also causes the polariton frequencies to blueshift. We refer to this mechanism as the exciton blueshift mechanism. Secondly, at higher exciton densities the exciton oscillator strength is reduced. This reduces the Rabi splitting and causes the lower polariton

branch states to shift nearer to the purely photonic dispersion, which lies at higher energies. Thus this second mechanism also causes a polariton blueshift. We refer to this as the Rabi quenching mechanism. The exciton blueshift and reduction in Rabi splitting per unit exciton density per QW are proportional to the interaction constants $g_{XX}$ and $\beta_X$ respectively. Using the GaN QW exciton Bohr radius[40] these may be calculated[39] as $g_{XX}$ ~0.85 µeV µm$^2$ and $\beta_X \sim -1.3$ µeV µm$^2$ (See Supplementary Discussion 8). Considering the LPB frequency blueshift due to $g_{XX}$ and $\beta_X$ and accounting for the 22 QWs in our system we obtain an effective nonlinear refractive index $n_2 = (3.3 \pm 0.9) \times 10^{-13}$ and $(6.5 \pm 1.5) \times 10^{-13}$ cm$^2$ W$^{-1}$ at $T = 100$ K and 200 K, respectively (Supplementary Discussion 8). These are within a factor of 1.8 of the values deduced from the experimental SPM spectral broadening. Thus the experiment and the first-principles interaction constants $g_{XX}$ and $\beta_X$ are in good agreement.

The interaction constants are comparable with those predicted and measured in GaAs ($g_{XX} \sim |\beta_X| \sim 2.5$ µeV µm$^2$ for linear polarisation)[39], when one accounts for the much smaller exciton Bohr radius of GaN[41]. They are also similar to interaction constants measured for trions or polarons in transition metal dichalcogenides (TMDs)[42,43]. The total nonlinearity ($g_{XX}$ and $\beta_X$) for a single QW is comparable with that observed per inorganic layer in hybrid inorganic-organic perovskites at around 516 nm[27]. In our AlInGaN device we access the UV spectral range and the nonlinearity is not restricted to cryogenic temperatures, as in the GaAs and TMD devices measured so far, and does not suffer from photo-bleaching as in many organic systems.

Finally, we note that from these microscopic considerations we expect the contributions of exciton blueshift and Rabi quenching to the polariton nonlinearity to be of the same order of magnitude. For the detunings in Fig. 5 we calculate that Rabi quenching contribution will be 2.4–2.8 times larger. From the point of view of using the waveguide for nonlinear optics it is not important to distinguish between the microscopic mechanisms since they produce qualitatively the same pulse modulations (see Supplementary Figs. 4 and 5d and Supplementary Discussion 5).

## Discussion

We have observed a strong power-dependent broadening of the spectra of optical pulses propagating in our polariton waveguide. We identify nonlinear pulse self-modulation as the mechanism for the following reasons. At the lowest powers the variation of spectral shape with power matches that expected for the well known self-phase-modulation mechanism[1]. The value of non-linearity we deduce assuming an SPM mechanism agrees within a factor of 2 with the one deduced from independently known microscopic properties of QW excitons. Furthermore, numerical evolution of the polariton propagation equations provides quantitative agreement with the spectral widths at low to intermediate powers and qualitative agreement at high powers. These agreements can only be expected if a polaritonic nonlinearity is really the mechanism responsible for the observed broadening. The numerical simulations also confirm that the spectral broadening corresponds to the temporal envelope of the pulses becoming strongly modulated.

Since we use a semiconductor system it is important to note that our spectra cannot be generated by thermal relaxation from higher energy electronic states resulting in broad luminescence spectra. We observe spectra which are initially narrow and broaden strongly with excitation power. The initial broadening is always symmetric about the pulse central frequency even as the pulse is detuned by different amounts compared to exciton frequency. By contrast, the PL spectra (see Fig. 2b) have a very different characteristic shape which is broad even at low power

and is naturally fixed to the electronic structure of the QWs. The strong power dependence observed for the resonant pulses was not observed and is not expected in PL. Furthermore, in our resonant excitation scheme, we directly excite the polaritons lying below the QW exciton transition. We, therefore, expect that, compared to the directly injected pulse intensity, there will be a negligible population of hot carriers which could create lumi-nescence. We finally note that the generated spectra have a complicated spectral structure, in particular the dependence of wavelength on position (see e.g. the highest power spectrum in Fig. 3b). By contrast, hot carrier relaxation is expected to lead to a smooth dependence on $y$ since the many electronic collisional interactions tend to populate all in-plane directions equally and incoherently.

We now compare our measured value of nonlinear refractive index $n_2$ to that in other relevant systems. We measure $n_2 = (1.9 \pm 0.3) \times 10^{-13}$ cm$^2$ W$^{-1}$ for 100 K and $n_2 = (3.7 \pm 1.0) \times 10^{-13}$ cm$^2$ W$^{-1}$ for 200 K using propagating sub-picosecond pulses. Table 1 summarises $n_2$ for a range of materials used for UV nonlinear optics as well as silicon nitride (which is of general importance in nonlinear optics), AlN and bulk GaN. Our value is three orders of magnitude larger than in the majority of these materials.

Values quoted for GaN are typically of order $10^{-14}$ cm$^2$ W$^{-1}$ for the near-infrared to visible spectral region[44,45] but values greater than $10^{-12}$ cm$^2$ W$^{-1}$ have been reported in the UV close to the band gap[46]. These results are orders of magnitude larger than can be expected from the well known material-independent scaling of $n_2$ with frequency[44,47]. This might be explained because they were measured using spatial nonlinear refraction at the pump spot position, which can be dominated by thermal effects or photo-generated free-carriers[45,48]. Since such effects have long relaxation times[48] it is not clear how useful they can be for ultrafast pulse modulations. In our work, by contrast, we measure modulations of the temporal envelope of propagating pulses. Even if free carriers are generated in our device their density could not vary with the sub-picosecond temporal shape of the pulse and so they cannot contribute to the SPM spectra we observe. We find $n_2$ values 1–2 orders of magnitude larger than those found in bulk GaN at visible wavelengths and attribute the large values to the strong photon-exciton coupling, which is well known to enhance nonlinearity[28]. To our knowledge, this is the first time the UV nonlinear refractive index in a GaN-based device has been measured using such a propagating pulse tem-poral modulation.

Having demonstrated a strong nonlinear response in the UV it is interesting to briefly consider some perspectives for how it could be used. One interesting direction could be nonlinear compression[3] to produce ultra-short broadband UV pulses. Here, pulses are spectrally broadened using nonlinear processes and then temporally compressed to produce shorter transform limited pulses. The compression may be achieved either with external dispersive elements or through temporal soliton dynamics[4]. Temporal solitons in polariton waveguides have already been demonstrated in the infra-red[28]. In refs. [10,13] short UV pulses are employed for transient absorption spectroscopy of organic molecules. Our spectral bandwidth of order 100 meV is already sufficient to resolve individual transitions in a variety of mole-cules such as, for example, pyrene[13] and atmospheric gases NO and HONO[12]. Broadband and short pulses may also be used for Coherent anti-Stokes Raman scattering (CARS). This technique can be enhanced when the pump pulses are resonant with elec-tronic transition of the medium under study, which for many chemical and biological samples are in the UV[15]. CARS sources can be used with microfluidic chips[49], and nonlinear pulse sources for CARS are being developed in silicon nitride in the

IR[50]. The required bandwidth is governed by the vibrational spectrum of interest and is often of order 100 meV. Required pulse energies are in the pico-Joule to nano-Joule range[15,49,50].

Closely related to soliton formation is the concept of dissipative soliton Kerr frequency combs[5]. These are high repetition rate trains of ultrafast pulses arising due to solitons propagating round an on-chip micro-resonator. They can be generated starting from CW light which, combined with GaN laser diode technology, raises the possibility of a fully on-chip integrated UV femtosecond source. UV frequency combs have important applications in spectroscopy of trapped ions. In ref. [51], for example, the authors use a ~100 meV broad, 2 mW (25 pJ per pulse) comb at 393 nm for spectroscopy of calcium ions. In ref. [16] trapped magnesium ions are Doppler cooled by a UV frequency comb with spectral width ~60 meV and average power 40–80 micro Watts (0.1 to 0.2 pJ pulses). Femtosecond UV pulses can also be used as pumps for single photon sources based on spontaneous parametric downconversion[20]. Since AlInGaN materials also have a strong second order nonlinear response[52] there is again the possibility of fully integrated sources. These various perspectives naturally span a range of UV wavelengths. An advantage of AlInGaN QWs is that the band edge is widely tuneable using parameters such as well width and material composition. Thus the operating wavelength of devices can be customised for particular applications.

We finally discuss the bandwidth over which our system can support the giant nonlinearity. The light-matter coupling which provides the strong interactions is a resonant effect and so the nonlinear bandwidth is ultimately limited to a few times the Rabi splitting[33], which is of order 100 meV in this device. While this does not match octave spanning supercontinuum (SC) sources in optical fibres[4] or highly optimised on-chip waveguides[6,22] our nonlinear bandwidth is already compatible with a range of applications. It is also important to note that the above mentioned SC sources typically use infra-red pumps and rely on nonlinear processes such as dispersive wave emission[6] and/or harmonic generation[22] to extend the spectrum into the UV. Such processes require millimetre long device or micro- to milli-Joule pump pulses, or both. Here we show a nonlinear effect which acts directly on nano-Joule UV pump pulses in only 100 μm propagation. The trade-off between bandwidth and nonlinear strength can also be engineered by adding more QWs to increase Rabi splitting.

In summary, we have reported on experimental observation of nonlinear self-modulation of UV pulses in an AlInGaN-based waveguide, in a polaritonic system, and in a sub-millimetre on-chip device. We take polariton nonlinearities into the UV in a robust, well developed material system which operates up to room temperature and does not suffer from photo-bleaching. The nonlinearities are orders of magnitude larger than in commonly used UV nonlinear materials over bandwidths compatible with a wide range of potential of applications. Our results, therefore, have potential to establish GaN polariton waveguides as a technological platform for ultrafast nonlinear optics without cryogenics in the technologically important UV spectral range.

## Methods

**Optical measurement setup.** Light was injected and collected through a single microscope objective by using a beamsplitter cube. Collected light was sent to an imaging spectrometer using confocal relay lenses allowing us to measure wavelength vs. either $y$ or angle. The spectrometer resolution depended on the width of its input slit. For the SPM measurements (Fig. 5) the slit was set to 50 μm width and the resolution was 1.7 meV, well below the initial pulse spectral FWHM of 4.2 meV.

**Grating coupling.** The gratings for light input and output were realized with a 20 nm-thick nickel metallisation beneath the SiO₂ cladding. They have a 130 nm periodicity with 65 nm metal stripes and were fabricated using electron-beam

lithography, thermal evaporation of the metal and lift-off. For output coupling of light the gratings diffract each wavelength out at a different angle corresponding to the wavenumber of the light in the waveguide (see Fig. 2c, d). We can see from Fig. 2c, d that the gratings can out-couple light from at least 347 nm to 365 nm corresponding to a bandwidth of at least 170 meV. This is limited by the width of the PL spectra rather than the gratings. For in-coupling of pulses the bandwidth is limited by the overlap of the waveguide dispersion and the incident pulse frequency-angle spectrum and in practice is given by the finite lifetime on the grating. We measure a decay length of ~3.5 μm on the grating, which corresponds to a ~0.6 ps lifetime and bandwidth of ~11 meV. To maximise coupling the central wavenumber of the incident light should match the polariton dispersion. This typically occurs between 0° and 20° depending on the wavelength (see Fig. 2c, d). We achieved optimal coupling by adjusting the incident angle of the light while monitoring the intensity at the output (see Supplementary Discussion 4). The angle must be adjusted to within ±1 degree of the optimal value to achieve good coupling, corresponding to the angular width of the modes seen in Fig. 2c, d. We note that gratings with different periodicity can be fabricated to work conveniently at a required wavelength and angle.

**Photoluminescence measurements.** For the PL measurements the excitation was at a spatial position between two gratings. The excitation beam was near to zero incidence angle (parallel to $z$—see Fig. 1a) and at a wavelength of 325 nm, much higher in energy than the QW band edge (see Fig. 1b). In this situation an electron-hole plasma is generated at the position of the excitation spot. Energy and momentum relaxation of the hot carriers populates the exciton and polariton states. Exciton PL within the light cone is emitted directly at the excitation spot. The polaritons are able to propagate away from the excitation spot and reach the output gratings where they are diffracted out and can be detected. No grating is needed at the excitation spot position for these PL measurements since the hot carriers can be excited near to zero in-plane momentum. For the exciton spectra in Fig. 2a PL was excited by 20 nJ femtosecond laser pulses except in the case of the curve labelled '4K CW', for which a 75 micro-Watt CW HeCd laser was used. For the polariton spectra in Fig. 2b the PL was excited by CW HeCd laser and excitation power was 2.5 mW in all cases apart from the curve labelled '4 K 0.25 mW'. For the CW HeCd laser the excitation spot was elliptical with FWHM 7 μm × 11 μm. For the femtosecond pulsed excitation the repetition rate was 1 kHz and the spot was elliptical with FWHM 11 μm × 16 μm.

An imaging spectrometer was used to measure either the spatial or angular distribution of the PL spectra. Spatial resolution allowed separation of the polariton emission from the gratings (shown in Fig. 2b) from the exciton emission at the excitation spot. For angular resolution we projected the Fourier plane of the collection objective onto the entrance slit of the spectrometer and measured $\lambda$ vs. angle from the $z$-axis towards the $x$-axis (see Fig. 1a). For these measurements the spectrometer resolution was 6.7 meV owing to the 200 μm entrance slit size used to collect enough light from the sample. To make the polariton modes more visible the exciton luminescence has been subtracted using spectra recorded in a polarisation transverse to the polaritons. The LPB emission angle vs. $\lambda$ was fit using a coupled oscillator model. The waveguide photonic dispersion, used as input for the fit, was calculated from an electromagnetic model (see Supplementary Discussion 1). The fitting parameters were the exciton frequency, the Rabi splitting, and a rigid frequency offset of the calculated photon dispersion which accounts for possible variations between nominal and actual layer thickness and refractive indexes. Including all three was essential to obtain a good quality fit. Further details of the angle-resolved PL measurements and dispersion fitting are given in Supplementary Discussion 3.

**Pulse propagation and modulation measurements.** For the pulse propagation experiments the resonantly injected laser pulses were generated using the frequency-quadrupled output of a tuneable optical parametric amplifier (OPA). The OPA was pumped at 800 nm by a 100-fs Ti:Sapphire regenerative amplifier with 1 kHz repetition rate. The pulses were spectrally filtered using a diffractive pulse shaper to be within the bandwidth of the grating couplers (see Supplementary Discussion 4). The pulse energy was adjusted using neutral density filters. After propagation the light was scattered out by a grating coupler and we measured $\lambda$ vs. $y$. Maximum input coupling was achieved by adjusting the excitation beam polarisation and position and wavenumber in the $x$, $y$ plane (see Fig. 1) to maximise the power at the output coupler.

**Deduction of nonlinear refractive index.** The effective nonlinear refractive index is given by ref. [1],

$$n_2 = \frac{\partial \phi}{\partial E_{\text{pulse}}} T_0 \frac{\lambda_0}{2\pi} \frac{A_{\text{eff}}}{L_{\text{loss}}\left[1 - \exp\left(-L/L_{\text{loss}}\right)\right]} \quad (1)$$

where $\partial \phi / \partial E_{\text{pulse}}$ is the rate of change of nonlinear phase with pulse energy obtained from the SPM fitting. $T_0$ is the pulse temporal width, $\lambda_0$ is the vacuum wavelength at the pulse center. $A_{\text{eff}} = 1$ μm is the waveguide effective nonlinear cross-sectional area using the standard formula from ref. [1]. $L_{\text{eff}} = L_{\text{loss}}\left[1 - \exp\left(-L/L_{\text{loss}}\right)\right]$ is the effective length over which the nonlinearity acts accounting for losses[1]. $L$ is the device length and $L_{\text{loss}}$ is the characteristic decay length due to absorptive losses.

The pulse width $T_0 < 430 \pm 40$ fs (independent of temperature) was obtained from the measured spectrum at low power under the assumption that the pulses are unchirped and Gaussian. The values of $\partial\phi/\partial E_{pulse}$ were obtained by fitting the experimental spectra with theoretical spectra of SPM-broadened Gaussian pulses[1] as described in the text. Further details are given in Supplementary Discussion 6.

Deduction of $n_2$ requires accurate deduction of the pulse energy $E_{pulse}$ inside the waveguide, which is proportional to the measured incident pulse energy and the coupling efficiency of the input grating coupler. We calculated the coupling efficiency using FDTD simulations (Lumerical 3D FDTD solver) and obtained between $5 \pm 1\%$ and $4.2 \pm 0.9\%$ depending on temperature. The uncertainties come from variations in material parameters obtained from different sources in the literature. We minimised the uncertainty in our values of $n_2$ by complementing the FDTD calculations with direct measurements of the output power from the waveguide vs. incident power, which fixes a relationship between coupling efficiency and losses. This strongly constrains the product of $E_{pulse}$ and $L_{loss}[1 - \exp(-L/L_{loss})]$ appearing in Eq. (1), and so reduces the dependence on $n_2$ the coupling efficiency. We note that we rigorously propagated statistical uncertainties in all model parameters to the final quoted values of $n_2$. Further detail is given in Supplementary Discussions 6 and 7. As well as the statistical uncertainty, disorder in the fabricated grating couplers could lead to lower coupling efficiency than that obtained via FDTD, while our assumption of unchirped Gaussian pulses is equivalent to assuming the minimum possible $T_0$ for our measured spectrum. Thus there may be a small systematic underestimation in our values of $n_2$. We finally note that throughout the paper we quote the coupled pulse energy which is the product of the measured incident pulse energy and the coupling efficiency.

## Data availability

The data supporting the findings of this study are freely available in the University of Sheffield repository with the identifier https://doi.org/10.15131/shef.data.13516574.

## Code availability

The custom codes used in this study are available from the corresponding author upon reasonable request.

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

## Acknowledgements

This work was supported by the Engineering and Physical Sciences Research Council (Grant Nos EP/R007977/1 and EP/N031776/1) and the Swiss National Science Foundation (grant number 200020_162657). The authors acknowledge funding from the Ministry of Education and Science of the Russian Federation through Megagrant No. 14.Y26.31.0015.

## Author contributions

P.M.W., R.B. and D.N.K. conceived and designed the experiments; D.M.D.P., P.M.W. and R.P.A.E. performed the experiments; J.-F. C., J.C., R.B. and N.G. conceived the sample design and contributed to its linear optical characterization. J.-F.C. grew the sample; Z.Z. fabricated the grating couplers; A.Y. performed numerical modelling and simulation of nonlinear pulse propagation; A.Y. and I.S. contributed to the design of the experiments. P.M.W performed electromagnetic simulations, polariton dispersion fitting and SPM model fitting; P.M.W., D.M.D.P. and A.Y. wrote the manuscript and Supplementary discussions with contributions from D.N.K., M.S.S., R.B., J.C. and J.-F.C; all authors contributed to the discussion and analysis of the data.

## Competing interests

The authors declare no competing interests.
