## [Peer Review File · Nature Communications]

REVIEWER COMMENTS

Reviewer #1 (Remarks to the Author):

Ultrafast-nonlinear ultraviolet pulse modulation in an AlInGaN polariton waveguide operating up to room temperature

The authors have reported on experimental observation of nonlinear self-modulation of UV pulses in an AlInGaN-based waveguide, in a polaritonic system, and in a sub-millimeter on-chip device.

It has been well known that QWs structures significantly increase nonlinear properties of semiconductor materials [for example Phys Rev Lett 62(9), p.1041, 1989] compared to bulk material. It is also not clear why cryogenic temperatures are required to exploit the nonlinear phenomenon. However, the novelty here is AlInGaN materials and research in the UV wavelength range.

I am quite surprised that the authors did not present photo-luminescence spectra of the investigated waveguide. I think luminescence properties are also very important to understand nonlinear behaviour of the device, particularly the comparison between CW and pulse pumping.

I didn't find any information about parameters of input and output gratings, but think if gratings were developed for efficient coupling of 200nm wavelength (a frequency quadrupled optical parametric amplifier pumped by a 800nm Ti:Sapphire laser), they would be very inefficient for a 325nm laser beam (HeCd laser).

The work contains comprehensive experimental work and theoretical analysis. The main idea of nonlinear self-modulation of UV pulses is original but needs further experimental confirmation. It is also not clear how self-modulation of UV pulses can be used in practice.

Conclusion: The manuscript is interesting and deserves to be discussed. The manuscript might be accepted for publication in Nature Communications after revisions are made.

Reviewer #2 (Remarks to the Author):

This is quite an interesting manuscript and can be published in this Journal. However, some clarifications are needed to help better understand the flow of the story:

1- I highly suggest some sections to be re-written for the sake of clarity. Even at the Introduction paragraph that needs to be clear, there are long and confusing sentences like this:

"For long-term applications it is important that the nonlinearity is effective without cryogenic cooling and has large magnitude so that it can be exploited with low power pulses on length scales suitable for eventual on-chip integration with other optical components."

2- Do you have a luminescence spectra of the device under study to add to the manuscript or to the Supplementary section?

3- You use Nickel grating coupler which likely has plasmon resonances at those wavelength enhanced by the grating. And your experiment shows strong non-linearity measured over short length of 100 micron. Do you have evidence that no nonlinear contribution comes from the plasmonic resonance of the grating coupler?

4- The manuscript says in multiple places "commonly known UV materials" without mentioning even one example name. I see you have cited some papers. But it's very helpful to mention a few of these UV materials that are very popular.

5- Please provide some refractive index information for the device layers.

6- Another note is that, a lot of the emphasis is on pulse measurement data and less on the device by itself, or why this specific device design. If the device is the core of this manuscript, then its importance and its properties (optical mode, electronics, etc) need to be further highlighted.

Reviewer #3 (Remarks to the Author):

This is an extension of the previous work of this group on spectral broadening of pulses in non-linear polaritonic waveguides [Ref. 23] to a new exciton-hosting material, which allows to extend this technique to room-temperature operation and UV spectral range. Both of the latter represent a significant novel aspect of the work since this regime is important for potential applications. In general, I find the manuscript well-written and would recommend publication. However, there are a few points that, in my opinion, should be further clarified before publication:

1. Since the authors make a strong statement about deployability of this device for biosensing and other applications, it would be good to learn more about the device limitations and performance benchmarking. In particular:

- a). How large is the bandwidth in which the device is operating (limited, of course, by the grating couplers)?
- b). What is the requirement on the angle of injection for the coming pulses?
- c). How does the device performance compare with the spectral broadening achieved using more traditional optical waveguides and photonics crystal fibres?

2. Based on numerical modelling of a system of coupled equations representing a photon and an exciton (presented in Supplementary Discussion 4), the authors conclude that the mechanism for spectral broadening of a resonantly transmitted pulse is the nonlinearity arising from the dependence of the Rabi frequency on the density of excitons. It is not clear how and why this mechanism is different from the the bleaching of Rabi coupling and effective blueshift that translates to the Kerr-like "saturation" nonlinearity referred to in Supplementary Discussion 8.

3. Presumably, the dependence of the Rabi splitting on the excitation power (proportional to exciton density) can actually be measured by the method described in Supplementary Discussion 2, and used to quantify the validity of the model?

4. In Ref. 23, the authors observed very similar spectral broadening without invoking the dependence of Rabi splitting on the exciton density. Is there a simple physical reason for why this mechanism becomes critical at room temperature?

5. The authors make a point that the "proper" photon dispersion including higher-order terms should be taken into account in order to observe the effects resulting from the interplay of dispersion and nonlinearity. How well can the coefficients of the expansion be determined from fitting to the experimental data?

6. Generally, the slowly-varying approximation for a pulse would break down when the pulse is compressed to within a few temporal oscillations of the field. By referring to this effect as one possible explanation of the observed discrepancy between experiment and numerics, do the authors imply that this really the case for their experiment? What exactly is the measured width of the output pulse for the largest input power?

Reviewer #1 (Remarks to the Author):

The authors have reported on experimental observation of nonlinear self-modulation of UV pulses in an AlInGaN-based waveguide, in a polaritonic system, and in a sub-millimeter on-chip device.

It has been well known that QWs structures significantly increase nonlinear properties of semiconductor materials [for example Phys Rev Lett 62(9), p.1041, 1989] compared to bulk material.

We agree that it is well known that proximity to QW optical resonances can enhance various kind of nonlinearity, such as in [Phys Rev Lett 62(9), p.1041, 1989] where enhanced second order nonlinearity is used for second harmonic generation.

Here, we focus on **third** order nonlinearities due to exciton-polaritons, which are 3 orders of magnitude larger than those in weakly coupled semiconductors, enabling nonlinear pulse modulation to be achieved in photonic chip geometries. To the best of our knowledge this has so far only been observed (by us) in a GaAs polariton system in the near infra-red at 5 K. In this respect the fact that we observe the 3rd order nonlinearities in UV in GaN polariton system at high T provides a significant novelty with respect to the previous studies. We now discuss these points more clearly in our revised introduction.

It is also not clear why cryogenic temperatures are required to exploit the nonlinear phenomenon.

As we show in this work, cryogenic temperatures are not needed in the case of our GaN sample. Most previous work on optical pulse nonlinearities using strong coupling between excitons and photons (polaritons) has been done with GaAs and shallow InGaAs quantum wells. The excitons in such wells have low binding energy so that cryogenic temperatures are needed to prevent the thermal energy from dissociating the excitons. In GaN the binding energy is much larger so we observe the polariton nonlinearity up to room temperature as well as in the UV spectral range. We now clarify this point towards the end of introduction.

However, the novelty here is AlInGaN materials and research in the UV wavelength range.

We agree that the AlInGaN materials and UV wavelength range are essential aspects of our work. We would like to note that the main focus of this paper is on the strong nonlinear effects, which were not previously demonstrated. The linear properties were already reported in [J. Ciers et. al., Phys. Rev. Appl. 7 034019 (2017)]. The novelty in our manuscript is that we demonstrate a polariton-based nonlinear modulation in the UV spectral range. To our knowledge this is also the first demonstration of third-order-nonlinear UV pulse modulation in a III-nitride-based waveguide. We have now clarified this point and tried to bring out the importance of these nonlinear properties in a re-written introduction and modified abstract.

I am quite surprised that the authors did not present photo-luminescence spectra of the investigated waveguide. I think luminescence properties are also very important to understand nonlinear behaviour of the device, particularly the comparison between CW and pulse pumping.

In the previous version we only presented two angle-resolved photoluminescence spectra of the polariton states and the discussion in the main text was very brief. We have now added additional PL spectra and significantly expanded the discussion of the PL data with sections in the main text and supplementary information. We have also added a comparison between the nonlinearly-generated broadened spectra and the PL spectra and described in detail the difference between the pulsed resonant pumping scheme and the above-bandgap pumping used to generate PL. Essentially, the PL spectra are always broad at any power since they have their origin in hot carrier relaxation. In contrast, the resonantly injected pulses start with the relatively narrower spectrum of the incident laser and then broaden rapidly with power due to nonlinearity. PL spectra are generated through thermal relaxation of hot carriers and so do not have the sub-picosecond timescales or coherence of pulses broadened through nonlinear-optical effects. We discuss the differences between nonlinear and PL spectra further below.

I didn't find any information about parameters of input and output gratings, but think if gratings were developed for efficient coupling of 200nm wavelength (a frequency quadrupled optical parametric amplifier pumped by a 800nm Ti:Sapphire laser), they would be very inefficient for a 325nm laser beam (HeCd laser).

Our gratings are suitable for the wavelengths of resonant excitation and PL emission (around 355 nm). We achieve 355 nm resonant excitation since the output of our parametric amplifier is tuneable, it is only pumped at 800 nm. For the non-resonant excitation of PL (using 325 nm) no grating is needed since we excite hot carriers at near zero in-plane wavenumber.

We have added parameters of the gratings (130nm periodicity, 50% duty cycle) to the methods section in the manuscript and clarified that the output of our optical parametric amplifier is tuneable. We have also added additional discussion clarifying the excitation mechanism for 325nm. As well as additional discussion, we have added additional panels Figure 1 b and c to clarify the two different excitation schemes (resonant and non-resonant).

The work contains comprehensive experimental work and theoretical analysis. The main idea of nonlinear self-modulation of UV pulses is original but needs further experimental confirmation.

We present the following evidence for nonlinear self-modulation of UV pulses.

- 1) At the lowest powers the evolution of the spectral width follows that which is well known as the signature of self-phase-modulation (SPM), one of the most fundamental nonlinear effects in optics. By considering the inverse Fourier transform of the spectra it can be seen that the spectral reshaping corresponds to a modulation of the temporal envelope of the pulse in the time domain.
- 2) The nonlinear strength we deduce from the SPM mechanism agrees within a factor of 2 with that deduced from independently known microscopic properties of quantum well excitons.
- 3) At low and intermediate powers the broadening of the spectra agree quantitatively and qualitatively (respectively) with numerical simulations based on evolution of the equations for

nonlinear polariton propagation. This agreement can only be expected if the phenomenon we observe is indeed caused by nonlinear polariton propagation. The simulations also confirm that the spectral broadening corresponds to the temporal envelope of the pulses becoming strongly modulated.

4) A general principle in conventional nonlinear optics is that for optical pulses propagating in a waveguide only a nonlinear process can transfer energy from one frequency state to another and result in a broadened spectral shape, as we observe. Linear propagation does not couple frequency components. Frequency-dependent loss can narrow a spectrum by preferentially absorbing some spectral components but cannot cause spectral broadening and, moreover, is independent of power. Thus the strong power-dependent spectral broadening must be the signature of a nonlinear effect.

In our system we should however exclude a possible alternative explanation applicable to semiconductors, namely some kind of thermal relaxation from higher energy electronic states resulting in broad luminescence spectra. We have now added extra experimental data showing the spectra generated through luminescence and extra discussion comparing it with the spectra generated under our resonant pulsed excitation. The PL spectra are broad even at low powers, do not vary strongly with power, and their shape is fixed to the electronic structure of the sample. By contrast our nonlinear spectra are initially narrow, broaden strongly with increasing power and always broaden around the pump frequency irrespective of its detuning relative to the exciton frequency. Thus our observations are consistent with a nonlinear-optical broadening effect rather than luminescence.

5) The generated spectra have a complicated spectral structure, in particular the dependence of wavelength on transverse position 'y' (see Fig. 3a, highest power). By contrast hot carrier relaxation is expected to lead to a smooth dependence on 'y' and on wavelength since the many electronic collisional interactions populate all in-plane momentum states equally and incoherently.

We have now added a discussion section to the manuscript where we summarise this evidence in order to make it clearer to the reader. We hope that this, along with our additional PL data and comparison of the nonlinear and PL spectra provides sufficient experimental confirmation that the spectral broadening does indeed arise from nonlinear pulse modulation.

It is also not clear how self-modulation of UV pulses can be used in practice.

While our main aim is to demonstrate that the relevant nonlinearity exists and can be exploited in the UV we agree that it is useful for the reader to understand the directions such nonlinearities can potentially open. We have now re-written the introduction to more clearly bring out the uses for third order optical nonlinearities acting to modulate optical pulses, and how these can be important in the UV spectral range in particular. We have also added a section of discussion at the end of the manuscript where we expand on a selection of these perspectives.

Conclusion: The manuscript is interesting and deserves to be discussed. The manuscript might be accepted for publication in Nature Communications after revisions are made.

We thank the reviewer for their interest in our work. We hope that with the revisions we have made the manuscript is now suitable for publication in Nature Communications.

Reviewer #2 (Remarks to the Author):

This is quite an interesting manuscript and can be published in this Journal. However, some clarifications are needed to help better understand the flow of the story:

We thank the reviewer for their positive assessment. We have now added substantial additional material to clarify the points raised below, which we now respond to one by one.

1- I highly suggest some sections to be re-written for the sake of clarity. Even at the Introduction paragraph that needs to be clear, there are long and confusing sentences like this:

"For long-term applications it is important that the nonlinearity is effective without cryogenic cooling and has large magnitude so that it can be exploited with low power pulses on length scales suitable for eventual on-chip integration with other optical components."

We have re-written the manuscript, taking care to use simpler sentence structure. In particular the introduction has been completely re-written.

2- Do you have a luminescence spectra of the device under study to add to the manuscript or to the Supplementary section?

We previously showed only angle-resolved photo-luminescence (PL) spectra and provided only a very brief discussion in the main text. We have now added additional PL spectra and a significantly expanded discussion of the PL properties (please see also response to the question regarding PL spectra from Reviewer 1). In particular we highlight the differences between the spectra generated through PL and those generated through nonlinear modulation of resonantly injected pulses.

3- You use Nickel grating coupler which likely has plasmon resonances at those wavelength enhanced by the grating. And your experiment shows strong non-linearity measured over short length of 100 micron. Do you have evidence that no nonlinear contribution comes from the plasmonic resonance of the grating coupler?

The observed nonlinear effect was only present in a very narrow angular bandwidth corresponding to the exciton polariton dispersion seen in angle-resolved photoluminescence (PL), examples of which are given in Fig. 2c-d. The generated spectral width was observed to decrease significantly when the incidence angle of the light was adjusted by small amounts ($\pm \sim 2$ degrees) at fixed incident power, corresponding to strongly reduced coupling of the incident light to the guided mode. It is highly unlikely that a plasmon resonance would have this very strong sensitivity to incidence angle, or that it would happen to be correlated with the position of the exciton-polariton resonance seen in PL. Furthermore, we observed no resonances in the PL spectra obtained from the regions with gratings apart from those corresponding to the excitons and exciton polaritons. These PL spectra, which we now show in the manuscript, agree very closely with the spectra from [J. Ciers et al., Phys. Rev. Appl. 7 034019 (2017)] which used etched dielectric rather than metallic gratings. If plasmon resonances were present then they would have caused visible features in the luminescence coming from the underlying semiconductor material either by absorption or scattering of light into surface waves. Finally we note that surface plasmons (SP) and surface plasmon polaritons (SPP) of

the kind formed by metals and metal gratings on dielectric surfaces can only be excited by TM polarised light, with the optical magnetic field parallel to the film. By contrast we excite with TE polarised light, orthogonal to TM so that we cannot excite SPs or SPPs.

4- The manuscript says in multiple places "commonly known UV materials" without mentioning even one example name. I see you have cited some papers. But it's very helpful to mention a few of these UV materials that are very popular.

We have now added a table 'Table I' showing a comparison of the nonlinear refractive index of a wide range of materials including the ones most used in the UV and some other (such as silicon nitride) which are of high importance in the field of nonlinear optics. We now refer to this table in the text so that the reader can see the materials at a glance. We have also added an extra discussion section to the end of the manuscript where we compare our nonlinear refractive index to those of other materials.

5- Please provide some refractive index information for the device layers.

We have added a figure (now Supplementary Figure 1) and some extra discussion in Supplementary Discussion 1 where we give the refractive index values we used and explain their origin.

6- Another note is that, a lot of the emphasis is on pulse measurement data and less on the device by itself, or why this specific device design. If the device is the core of this manuscript, then its importance and its properties (optical mode, electronics, etc) need to be further highlighted.

The core of this manuscript is the very high nonlinearity in the UV spectral range which originates from the formation of exciton-polaritons, and which was not previously reported in the UV. We have now provided additional discussion of device linear properties since these help to understand the platform which is supporting this large nonlinearity. In particular we plot the waveguide optical mode dispersion relation and mode spatial distribution in Supplementary Figure 1, along with a discussion of how they were calculated in Supplementary Discussion 1. We have also added additional photoluminescence data and an extensive discussion to make the electronic structure of the quantum wells more clear. We hope that this additional discussion helps to place the nonlinear results in the proper context.

Reviewer #3 (Remarks to the Author):

This is an extension of the previous work of this group on spectral broadening of pulses in non-linear polaritonic waveguides [Ref. 23] to a new exciton-hosting material, which allows to extend this technique to room-temperature operation and UV spectral range. Both of the latter represent a significant novel aspect of the work since this regime is important for potential applications. In general, I find the manuscript well-written and would recommend publication. However, there are a few points that, in my opinion, should be further clarified before publication:

1. Since the authors make a strong statement about deployability of this device for biosensing and

other applications, it would be good to learn more about the device limitations and performance benchmarking. In particular:

As the reviewer notes above, the primary motivation of the paper is to demonstrate polariton nonlinearity acting on pulses in the UV and also up to room temperature. The statement previously made in the introduction was intended to provide some explanation to the reader for why nonlinearity in the UV spectral range is important and was not intended to be a strong statement about deployability for a particular application. Indeed our device is not optimised for any particular application and serves merely to prove the principle that polariton pulse modulation is possible in the UV. We have re-written the introduction to provide a more broad perspective on the importance of both third order temporal nonlinearities and of the UV spectral range. We also now consider some specific perspectives in the discussion section at the end of the paper, which includes a discussion of the fundamental limitations of the waveguide polariton approach. We respond to the particular questions in order:

a). How large is the bandwidth in which the device is operating (limited, of course, by the grating couplers)?

The grating couplers are not the main limit on the device bandwidth. The input coupling bandwidth is of course limited by the grating coupler (to about 11 meV full-width-at-half-maximum) since there is a phase matching condition between the guided mode and the incident light. However, this is easily sufficient to in-couple sub-picosecond pump pulses. There is no bandwidth limit on the out-coupled light – widely spaced wavelengths will simply be scattered out at widely spaced angles (see Fig 2 c,d). We have added a ‘Grating Coupling’ section in the ‘Methods’ part of the paper explaining these aspects.

A separate issue is the bandwidth over which the strong polariton nonlinearity operates. Fundamentally, the strong nonlinearity is obtained by strong coupling of photons and quantum well excitons. This is a resonant effect with width characterised by the Rabi splitting (~90meV for temperatures up to 200K). The resonance both enhances the nonlinearity but also limits its bandwidth to frequencies within a few Rabi splitting of the exciton resonance [P. M. Walker et. al., Light. Sci. Appl. 8, 6 (2019)]. This can be seen in Fig. 3h where the nonlinear broadening on the long wavelength side of the peak (far from the exciton) is weak for pulses tuned 100 meV away from the exciton. In practice one can engineer the tradeoff between nonlinear strength and bandwidth by varying the number of quantum wells in the waveguide. Increasing the number of wells increases the Rabi splitting and hence the bandwidth, but spreads the excitons among more wells so that the interactions are weaker.

We have now added text clarifying the importance of the nonlinear bandwidth in the discussion section at the end of the paper.

b). What is the requirement on the angle of injection for the coming pulses?

The injected light should be within +/- 1 degree of the angle corresponding to the guided mode wavenumber at the wavelength of interest. This can be obtained from the polariton dispersion relations shown in Fig. 2c-d and is in the range 0-20 degrees depending on wavelength. The grating periodicity can be chosen to couple a wavelength of interest at a convenient angle.

c). How does the device performance compare with the spectral broadening achieved using more traditional optical waveguides and photonics crystal fibres?

Compared with some optical fibre and waveguide supercontinuum sources the spectral broadening is relatively small (around 100meV compared to octave-spanning). On the other hand these alternative approaches use pump pulses in the infra-red and rely on dispersive wave emission or frequency up-conversion to transfer a small fraction of the power into the UV. They also typically require either several-millimeter long devices or micro- to milli-Joule pump pulses or both. Having a strong nonlinearity which operates directly on UV pulses is essential for application based on, for example, pulse compression, soliton propagation or dissipative-soliton Kerr frequency combs. We find strong nonlinearity acting in a 0.1 mm device for nano-Joule pulses. Apart from these considerations it is worth noting that our bandwidth is already compatible with a variety of time-resolved spectroscopy applications such as transient absorption or coherent anti-stokes Raman spectroscopy.

We have now added a section at the end of the manuscript where we discuss some of these perspectives and alternative approaches in order to better place both the fundamental nonlinearity and the spectral broadening we observe in a proper context.

2. Based on numerical modelling of a system of coupled equations representing a photon and an exciton (presented in Supplementary Discussion 4), the authors conclude that the mechanism for spectral broadening of a resonantly transmitted pulse is the nonlinearity arising from the dependence of the Rabi frequency on the density of excitons. It is not clear how and why this mechanism is different from the the bleaching of Rabi coupling and effective blueshift that translates to the Kerr-like “saturation” nonlinearity referred to in Supplementary Discussion 8.

We did not make the discussion of nonlinear mechanisms clear in the previous version of the manuscript. There are two microscopic mechanisms which contribute to the observed nonlinear response. Bleaching of the Rabi coupling with density is the same effect as dependence of Rabi frequency on density. Increasing intensity (exciton density) reduces the exciton oscillator strength and hence also the Rabi splitting. We have modified the text to use a single consistent terminology and call this effect ‘Rabi quenching’. In the second mechanism the exciton resonance frequency increases (blueshifts) with intensity. We now consistently call this ‘exciton blueshift’.

These two microscopic mechanisms have the same effect on the lower polariton branch states in which our pulses propagate. As shown in Supplementary Figures 4 and 5d both mechanisms lead to qualitatively the same observed spectra in the numerical simulations so that we cannot say which mechanism is dominant. The reason for this is that Rabi saturation causes the lower polariton branch (LPB) to shift towards the uncoupled photon mode at higher frequencies, causing a frequency blueshift of the polaritons. Meanwhile exciton blueshift also causes the LPB to blueshift. To lowest order these effects are indistinguishable. We can, however, estimate the fractional contributions of the two mechanisms to the observed nonlinearity from first principles consideration of the quantum well excitons. We do this in what is now Supplementary Discussion 8. We have also added extra discussion in a section ‘Comparison with nonlinearity from microscopic exciton properties’ in the main text to clarify these points.

3. Presumably, the dependence of the Rabi splitting on the excitation power (proportional to exciton density) can actually be measured by the method described in Supplementary Discussion

2, and used to quantify the validity of the model?

Unfortunately this is not easy to do for several reasons. First, the upper polariton branch is not visible so there is no way to separate the contributions of Rabi quenching and exciton blueshift to the observed polariton blueshift. Secondly, when exciting non-resonantly it is not trivial to deduce the number of particles in the system and so extract a nonlinear coefficient. Deducing the particle number is easier with propagating pulses where the incident and transmitted pulse energies can be directly measured. Thirdly, when considering positions directly under a nonresonant excitation spot it is possible that high free carrier densities can modify the refractive indexes of the layers leading to changes in the photonic mode and consequent polariton frequency shifts. While we expect the exciton-related nonlinearity to be dominant these effects would make the analysis too complicated to draw meaningful conclusions. We note that the free carrier effects are not relevant for our experiments where we inject pulses resonantly and they rapidly propagate away from the pump spot.

Although the above method cannot measure the Rabi quenching directly we do compare the nonlinearity measured from self phase modulation of the propagating pulses to the one deduced from the independently known microscopic properties of excitons and find good agreement (within a factor of 2). This verifies that the measured nonlinearity originates from excitonic interactions. The comparison is made in the manuscript in the section “Comparison with nonlinearity from microscopic exciton properties”. Further details are given in supplementary discussion 8.

4. In Ref. 23, the authors observed very similar spectral broadening without invoking the dependence of Rabi splitting on the exciton density. Is there a simple physical reason for why this mechanism becomes critical at room temperature?

We do not believe the Rabi quenching mechanism depends on temperature. Both exciton blueshift and Rabi quenching mechanisms always contribute to the nonlinearity. In Ref. 23 (now Ref. 33) we used a simplified model with a single phenomenological exciton interaction constant (as we show in Supplementary Discussion 5 of this paper the two mechanisms produce the same pulse modulations, so can be combined) whereas in this paper we use a more detailed analysis which treats the contributions separately.

5. The authors make a point that the “proper” photon dispersion including higher-order terms should be taken into account in order to observe the effects resulting from the interplay of dispersion and nonlinearity. How well can the coefficients of the expansion be determined from fitting to the experimental data?

For the simulations to most accurately reflect the experimental device the essential point is that the single-particle polariton dispersion one obtains from the coupled photon and exciton equations matches the experimentally measured polariton dispersion, examples of which are shown in Figs. 2 c and d in the paper. As was discussed in [J. Ciers et. al., Phys. Rev. Appl. 7 034019 (2017)] the polariton dispersion arises from a combination of the Rabi splitting and the dispersion of the purely photonic modes. It is not possible to experimentally access the photon dispersion so it should be calculated from the known refractive indexes of the materials using an exact solution of Maxwell's equations. The overall agreement of the numerical and experimental polariton dispersions can then

be ensured by using this calculated photon dispersion as input when fitting the experimental data with a coupled oscillator model to extract the Rabi splitting.

We calculated the photon dispersion using the freely available CAMFR waveguide eigenmode solver. We have added additional discussion to Supplementary Discussion 1 to explain this in more detail. A new Supplementary Figure 1 gives more details about the refractive indexes, photon dispersion and guided mode profile in the growth direction. The procedure we used for fitting the experimental polariton dispersion is described in Supplementary Discussion 3. The expansion coefficients for the photon dispersion can be obtained from the eigenmode solutions with high accuracy. Combined with the Rabi splitting from the polariton fitting these give an accurate parametrisation of the overall polariton dispersion, which is plotted as solid white lines in Figs. 2 c and d. It can be seen that these curves agree well with the intensity maxima (the small discrepancy for LPB 2 in Fig. 2c is a nonlinear effect explained at the end of Supplementary Discussion 3).

6. Generally, the slowly-varying approximation for a pulse would break down when the pulse is compressed to within a few temporal oscillations of the field. By referring to this effect as one possible explanation of the observed discrepancy between experiment and numerics, do the authors imply that this really the case for their experiment? What exactly is the measured width of the output pulse for the largest input power?

We thank the reviewer for drawing our attention to this point. We are not able to directly measure the temporal width of the pulse due to its low power. However, we can assess the lower bound of the pulse width from the spectrum. The width 50 meV, where the numerical and experimental curves begin to separate, is about $1/70$ of the carrier frequency. Thus we are likely not in the few-optical-cycle regime where the approximation breaks down completely, although small numerical errors could still have a cumulative effect on the complicated dynamics at the highest powers. We have updated the manuscript at the end of the section on the numerical results to include these other possible mechanisms and make clear the scale of the pulse width compared to the carrier frequency.

REVIEWERS' COMMENTS

Reviewer #1 (Remarks to the Author):

I am happy with authors response and think manuscript can be accepted for publication.

Reviewer #2 (Remarks to the Author):

The authors have answered my questions, and have added the required materials into the manuscript. It's good to go.

Reviewer #3 (Remarks to the Author):

The authors have substantially revised the manuscript in response to the referees' comments. In particular, they have extended the discussion of the main mechanisms for the observed nonlinear effects (new Suppl. Discussion. 8), and significantly clarified their statements with regard to the applicability of the model.

I find their responses adequate, and the manuscript a much-improved version from the previous iteration. I therefore recommend the publication of this revised manuscript.